# Differential damage and repair of DNA-adducts induced by anti-cancer drug cisplatin across mouse organs

Askar Yimit[1], Ogun Adebali [2], Aziz Sancar[1,3] & Yuchao Jiang [3,4,5]

The platinum-based drug cisplatin is a widely used first-line therapy for several cancers. Cisplatin interacts with DNA mainly in the form of Pt-d(GpG) di-adduct, which stalls cell proliferation and activates DNA damage response. Although cisplatin shows a broad spectrum of anticancer activity, its utility is limited due to acquired drug resistance and toxicity to non-targeted tissues. Here, by integrating genome-wide high-throughput Damage-seq, XR-seq, and RNA-seq approaches, along with publicly available epigenomic data, we systematically study the genome-wide profiles of cisplatin damage formation and excision repair in mouse kidney, liver, lung and spleen. We find different DNA damage and repair spectra across mouse organs, which are associated with tissue-specific transcriptomic and epigenomic profiles. The framework and the multi-omics data we present here constitute an unbiased foundation for understanding the mechanisms of cellular response to cisplatin. Our approach should be applicable for studying drug resistance and for tailoring cancer chemotherapy regimens.

---

[1] Department of Biochemistry and Biophysics, School of Medicine, University of North Carolina at Chapel Hill, Chapel Hill, NC 27599, USA. [2] Molecular Biology, Genetics and Bioengineering Program, Faculty of Engineering and Natural Sciences, Sabanci University, Istanbul, Turkey 34956. [3] Lineberger Comprehensive Cancer Center, University of North Carolina at Chapel Hill, Chapel Hill, NC 27599, USA. [4] Department of Biostatistics, Gillings School of Global Public Health, University of North Carolina at Chapel Hill, Chapel Hill, NC 27599, USA. [5] Department of Genetics, School of Medicine, University of North Carolina at Chapel Hill, Chapel Hill, NC 27599, USA. Correspondence and requests for materials should be addressed to A.S. (email: aziz_sancar@med.unc.edu) or to Y.J. (email: yuchaoj@email.unc.edu)

Cisplatin, a platinum (Pt) coordination complex, is one of the most effective chemotherapeutic drugs used to treat several cancers, including testicular, ovarian, cervical, head, neck, non-small-cell lung cancer, and colorectal cancer[1–4]. Despite the fact that cisplatin can bind a wide range of cellular components, including proteins, RNA, membrane phospholipids, microfilaments, and thiol-containing peptides, DNA is considered a major target for cisplatin[2]. Once inside cells, cisplatin undergoes aquation, and the platinum atom of cisplatin binds covalently to the $N^7$ position of purines resulting in about 65% GpG, 25% ApG 1,2-intra-strand crosslinks, and ~5–10% GpNpG 1,3- intra-strand crosslinks, as well as a lower percentage of inter-strand crosslinks[5]. In response to cisplatin, cells activate multiple repair pathways, among which nucleotide excision repair pathway constitutes the main mechanism to detect and repair cisplatin-induced DNA adducts[6–8]. Two major nucleotide excision repair pathways, transcription-coupled repair (TCR) and global repair (GR), are well known to remove cisplatin-induced DNA adducts. TCR acts on the transcribed strands (TS) of active genes, while GR acts on the non-transcribed region of the genome, as well as the non-transcribed strands (NTS) of transcribed genes[9].

Although cisplatin shows a broad spectrum of anticancer activity, its utility is limited due to acquired drug resistance and serious side effects. Cisplatin resistance, which often results in disease recurrence, originates from multiple cellular self-defence adaptations, including reduced uptake and increased drug efflux, inactivation by proteins (e.g., metallothionein), small molecules (e.g., glutathione), and increased damage repair or tolerance[1,10]. In addition, common side effects associated with cisplatin treatment are ototoxicity, peripheral neuropathy, myelosuppression, and nephrotoxicity[11]. Another limitation in the use of cisplatin is damage to non-targeted tissues, suggesting that long-term off-target effects induced by the chemotherapeutic drugs are one of the major factors causing mortality in cancer survivors in later stage of life[12].

Since the discovery of cisplatin in the early 1960's, considerable efforts have been made to increase its anti-cancer drug efficiency and meanwhile to minimize its side effects to normal tissues[13]. A major barrier to a comprehensive understanding of the underlying molecular mechanism that related cisplatin-induced drug resistance and side effects is, however, a lack of approach that allows precise and high-resolution measurements of the genome-wide cisplatin-induced damage and repair in a high-throughput manner. Furthermore, most of the data from existing studies were generated using isolated cell lines, which can be misleading when extending the application to the in vivo experiments and clinical trials[14].

Here, we adopted high-throughput Damage-seq, eXcision Repair-seq (XR-seq), and RNA-seq to generate an integrated map of DNA damage, repair, and gene expression at single-nucleotide resolution across four mouse organs. Our experimental and analytical framework presented in this study serve as a resource for researchers interested in DNA damage and repair associated with cisplatin treatment in mouse models. Our analysis of the high-throughput data from the in vivo experiments shed lights upon not only the mechanisms of cisplatin-induced DNA-damage and repair, but also the cytotoxicity and drug resistance, both of which are important for chemotherapy regimens. The data we generated provide a platform for further research on optimizing cisplatin treatment efficacy and reducing side effects.

## Results

**Overview.** In this work, we present an experimental and analytical framework where we systematically assay and profile DNA damage, excision repair, and gene expression in a genome-wide fashion across four mouse organs. Figure 1a outlines the experimental design. Specifically, cisplatin was administered by an intraperitoneal injection in mice. DNA damage, excision repair, and gene expression were measured after 4 h cisplatin treatment by damage sequencing (Damage-seq)[15], excision repair sequencing (XR-seq)[16], and RNA sequencing (RNA-seq), respectively. All experiments were performed with two biological replicates. Genome-wide DNA damage, excision repair, and gene expression data show that samples from the same organs are clustered together by both principal component analysis (PCA) (Fig. 1b) and hierarchical clustering using the pairwise Euclidean distance between samples (Supplementary Figure 1). In addition, we retrieved publicly available epigenomic profiles across the four organs by ChIP-seq (H3K4me1, H3K4me3, H3K27ac, H3K27me3, H3K36me3, and POLR2A) and DNase-seq (DNase I hypersensitivity site) from the ENCODE Project. All of the ENCODE epigenetic data were generated from mice without cisplatin treatment, and represent overall organ-specific chromatin states, transcription factor binding, and histone modifications patterns[17]. Refer to Materials and Methods for details on experimental design and data processing. In the following sections, we describe how Damage-seq and XR-seq data are quantified and normalized, how gene expression is evaluated by RNA-seq, and then compare and contrast relationships between DNA damage, repair, RNA expression, and epigenetic markers.

**Cisplatin-induced DNA damage maps across mouse organs.** Here, we adopted a previously developed high-throughput sequencing protocol, Damage-seq[15], for assaying genome-wide DNA damage with single base pair resolution. Damage-seq has been successfully applied to map DNA damage caused by cisplatin[18] and ultraviolet radiation[15] onto the human genome in vitro. In this study, we applied Damage-seq to generate genomic profiles of cisplatin-induced DNA damage across four mouse organs (Supplementary Data 1). GG (Guanine-Guanine dinucleotide) is enriched 1–2 bp upstream of the 5′-end of the reads across all samples, indicating adequate data quality (Supplementary Figure 2). Notably, while we do not observe other dinucleotide enrichment, there is a preference for adenine 5′ to the GG dinucleotides of our Damage-seq reads (Supplementary Figure 3). Refer to Materials and Methods for details on library preparation, sequencing, and bioinformatic analysis.

Damage-seq measures the overall damage induced by cisplatin minus the accumulated excision repair from 0 to $T = 4$ h after cisplatin treatment by TCR in the TS and the GR in the TS and NTS:

$$\text{Damage}^{\text{TS}} = \sum{}^{\text{Damage}} - \int_0^T \text{GR}_t^{\text{TS}} \mathrm{d}t - \int_0^T \text{TCR}_t \mathrm{d}t,$$
$$\text{Damage}^{\text{NTS}} = \sum{}^{\text{Damage}} - \int_0^T \text{GR}_t^{\text{NTS}} \mathrm{d}t. \tag{1}$$

For data normalization, we first adjusted for the sequencing depth by dividing the total read counts by a sample-specific library size factor. Then, we adjusted for the number of GG dinucleotide sequences in each gene. We found that the number of GG dinucleotide sequences in the TS and NTS are highly correlated across all genes ($r = 0.99$, Supplementary Figure 4) and since cisplatin mainly induces Pt-d(GpG) damage, this achieves better normalization results than adjusting for the gene lengths, as shown in Supplementary Figure 5. After data normalization, we found that across all samples and organs, the damage level from the TS was lower than that from the NTS, due to ongoing TCR on the transcribed strand for 4 h until the

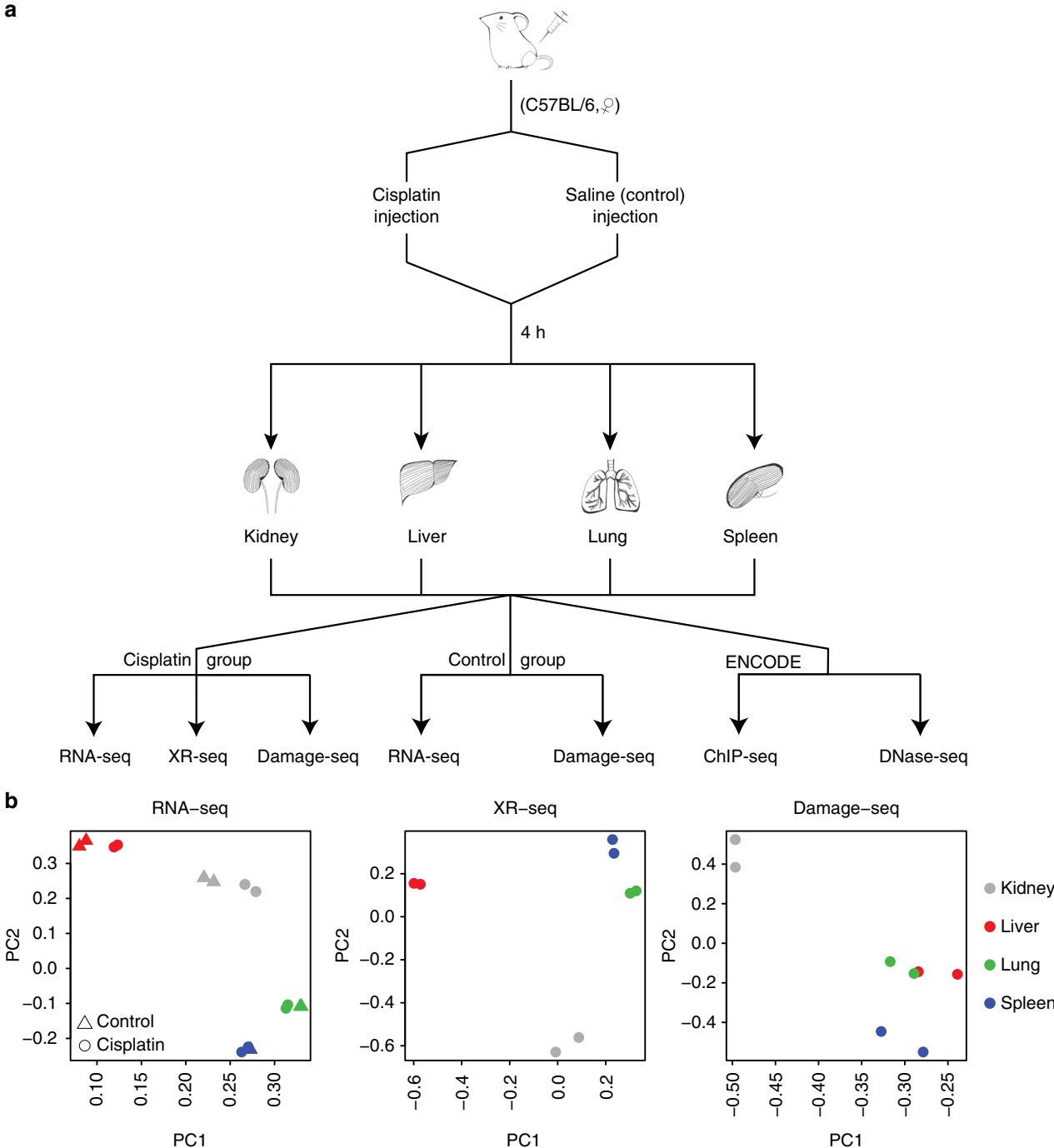

**Fig. 1** Overview of experimental design. **a** Outline of the experimental design. Cisplatin was administered by intraperitoneal injection in mouse. DNA damage, excision repair, and gene expression were assayed by Damage-seq, XR-seq, and RNA-seq. **b** PCA reveals tissue-specific damage, repair, and transcription across four mouse organs. All experiments were performed with two biological replicates

data were collected (Supplementary Figure 6). We then applied PCA to the normalized read count matrix from the NTS, where a negligible fraction of total DNA damage is repaired by GR only (Refer to Section "Effect of transcription on damage formation and repair" for more details). We found that samples from the same organ are clustered together, indicating organ-specific DNA damage at 4 h (Fig. 1b). This was further confirmed by hierarchical clustering on the pairwise Euclidean distance between the samples (Supplementary Figure 1c) using highly variable genes.

We further investigated the Pt-d(GpG) di-adduct formation efficiency across all organs both in silico and in vivo and found that cisplatin induces DNA damage at different rates, with kidney being the highest and spleen the lowest (Supplementary Figure 7a). For in silico inference, we first compared the NTS reads from Damage-seq for each sample to a pseudo reference constructed across all samples with adjustment of library size factor and then inferred the relative DNA damage. Our in vivo immuno-slot blot analysis, which detects unrepaired damage in the genome, confirmed that kidney is a major site of cisplatin

DNA-adduct formation, followed by liver and lung, while spleen is a minor site for cisplatin-induced DNA-adduct formation (Supplementary Figure 7b). These results are consistent with what has been reported in various studies of cisplatin DNA-adduct formation using different approaches in the rodent model system[19–21]. We further investigated the damage in mitochondrial DNA and found that in addition to nuclear DNA, mitochondrial DNA is a major target of cisplatin (Supplementary Figure 8), which is consistent with a previous report[22] and is possibly due to the lack of DNA excision repair in the mitochondria[23].

**Cisplatin-induced DNA repair maps across mouse organs**. To reconstruct genome-wide DNA repair maps, XR-seq libraries for cisplatin damage were prepared by adapting the XR-seq method previously developed[16]. Specifically, the 21- to 31- nucleotide-long oligomer products generated by nucleotide excision repair were captured by IP and ligated to adaptors on both ends. The ligation products were then purified by IP with the anti-cisplatin antibody, and the Pt adducts were reversed by incubation with NaCN. Finally, the oligomers were amplified to generate a library which was sequenced. XR-seq has been successfully adapted to assay genome-wide excision repair with single-nucleotide resolution in human[16], mouse[24], plant[25], bacteria[26], and yeast[27]. In this study, we applied XR-seq to quantify DNA excision repair in four mouse organs after cisplatin treatment (Supplementary Data 1), each with two biological replicates. Lengths of the excised oligomers fall within the range of the 21–31 nucleotide (nt) with a median of 26–27 nt (Supplementary Figure 9). As a target of cisplatin, GG dinucleotides are enriched 5–8 nt upstream of the 3′-end of the reads across all samples (Supplementary Figures 10 and 11), which is consistent with previous high-resolution study that mapped cisplatin damage and repair in cultured mammalian cells[18].

Unlike Damage-seq, XR-seq offers a snapshot of the ongoing excision repair (XR), which includes both TCR and GR on the transcribed strand and GR on the non-transcribed strand at time $T = 4h$:

$$\mathrm{XR}^{\mathrm{TS}} = \mathrm{GR}_T^{\mathrm{TS}} + \mathrm{TCR}_T^{\mathrm{TS}}$$
$$\mathrm{XR}^{\mathrm{NTS}} = \mathrm{GR}_T^{\mathrm{NTS}}. \tag{2}$$

In a similar fashion to Damage-seq, read counts from XR-seq were normalized to adjust for biases introduced by sample-specific sequencing depth and gene-specific number of GG dinucleotides. After data normalization, we found that, across all samples and organs, there is higher excision repair in the TS compared to the NTS, due to TCR specific to the TS (Supplementary Figure 12). PCA analysis on the normalized read count matrix also shows organ-specific excision repair patterns (Fig. 1b, Supplementary Figure 1b). For the mitochondrial DNA, we found on average 5.4% of the reads contain GG dinucleotide, compared to a proportion of 73.2% for the non-mitochondrial DNA (Supplementary Data 1). The greater amount of background noise in mitochondria implies a lack of excision repair in the mitochondrial DNA[23].

**Cisplatin-induced gene expression across mouse organs**. RNA-seq is fundamental for better understanding of transcriptomic dynamics in cells and tissues under different conditions[28,29]. To investigate the mechanistic link between cisplatin-induced DNA damage, repair, and gene transcription regulation and to idenfity cisplatin-induced gene expression patterns, we adopted RNA-seq to perform transcriptomic profiles of mouse organs with or without cisplatin treatment (Fig. 1a). We used reads per kilo base per million reads (RPKM) for data normalization, COMBAT[30] for batch correction, and DESeq2[31] for differential expression analysis.

We found 1475, 364, 1122, and 193 significantly up- and downregulated genes in kidney, liver, lung, and spleen, respectively (Supplementary Figure 13, Supplementary Data 2). Downregulated genes were more prevalent compared to upregulated genes, implying a cisplatin-related global transcription repression. Across all organs, 3 genes are significantly upregulated, while 12 genes are significantly downregulated (refer to Fig. 2 for Venn diagram of genes with significant differential expression across the four organs). Among the upregulated genes, *Cdkn1a* (P21) is a cyclin-dependent kinase inhibitor and a target gene of P53, which inhibits both cell cycle progression and replication under stress condition[32]; PER1 is a core circadian clock factor that plays essential role in the molecular clock mechanism[33]. Among the downregulated genes, DOCK1 and DENND1A proteins belong to small GTPase signal transduction regulatory pathway, while the rest play a role in fundamental cellular processes[34,35].

Cisplatin-induced DNA damage causes a variety of cellular responses (e.g., replication arrest, transcription inhibition, cell cycle arrest, DNA repair, and apoptosis) and activates multiple signal transduction pathways[36]. To gain insight into the functions of the differentially expressed genes, we carried out gene ontology (GO) analysis across mouse organs. Results from the GO analysis revealed that a large subset of the differentially expressed genes in kidney, liver, and lung after cisplatin treatment are primarily involved in fundamental cellular processes such as regulation of cell cycle, regulation of localization, metabolic process regulation, cellular response to external and endogenous stimuli, cellular communication and signaling, and subcellular component movement (Supplementary Figure 14). Interestingly, in addition to *Per1* being significantly upregulated across all organs, we found upregulated genes in kidney (*Csnk1e, Atf4, Hnrnpu, Bhlhe40, Ddx5, Nfil3, Id1, Crem, Noct, Impdh2, Mybbp1a, Csnk1d, Sirt1, Bhlhe41, Klf10, Cdk5r1, Sik1, Adamts1, Crem, Egr1 and Adrm1*) and lung (*Ddx5, Id1, Fas, Ciart, Sirt1, Klf9, Sik1, Adamts1,* and *Fzd4*) were enriched in circadian rhythm and rhythmic process in GO annotation. We did not find statistically significant enrichment of genes listed under circadian rhythm in GO annotation in liver or spleen. However, in addition to *Per1*, both *Klf10* and *Klf9* also demonstrated increased expression in liver, while *Sik1* and *Klf9* exhibited increased expression in spleen. Our results agree with previous findings that cellular DNA damage response is closely linked to circadian clock regulation[21,37].

To identify biological signaling pathways that are regulated after cisplatin treatment, we further carried out Ingenuity Pathway Analysis (IPA) on significantly up- and downregulated genes (adjusted *t*-test *p*-value less than 0.05 and absolute log fold change >2). We identified significantly regulated pathways across organs, as reported in Supplementary Data 3, and found that the ATM (ataxia-telangiectasia mutated protein) signaling pathway was among the most significantly upregulated pathways in kidney, liver, and lung. This result is consistent with our finding that p53 signaling-related genes were enriched in kidney, liver, and lung in our GO term analysis (Supplementary Figure 14). In addition, protein kinase A (PKA) signaling, synaptic long-term depression, and cardiac hypertrophy signaling pathways were downregulated across all organs. The PKA signaling pathway mediates diverse cellular processes, including growth, development, memory, and metabolism. Interestingly, it has been shown that inactivation of PKA signaling results in increased DNA repair and resistance to cisplatin[38].

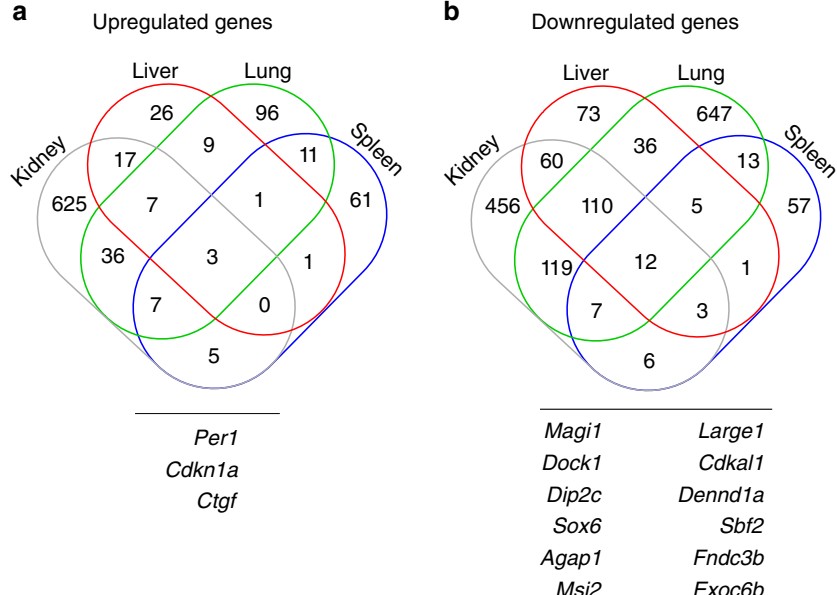

**Fig. 2** Cisplatin-induced differential gene expression between treated and control groups. Venn diagram shows the numbers of significantly differentially expressed genes across mouse kidney, liver, lung, and spleen four hours after treatment with cisplatin. **a** Three genes are significantly upregulated. **b** Twelve genes are significantly downregulated in all mouse organs

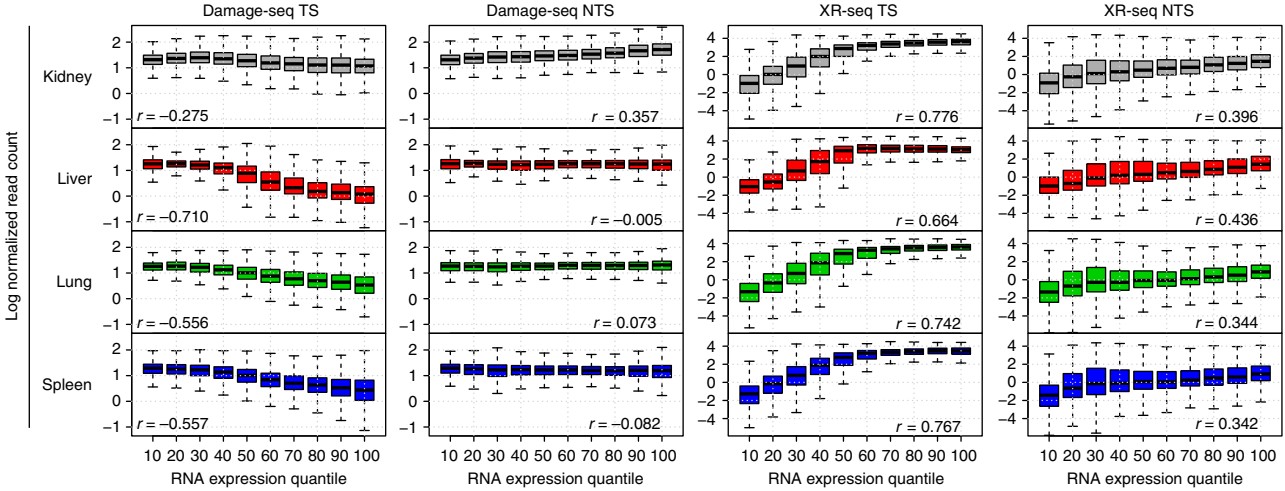

**Fig. 3** Effect of transcription on cisplatin-induced damage formation and repair. Genes are grouped based on their gene expression quantiles (*x*-axis) in the cisplatin-treated group. Log normalized read counts for DNA damage and repair (*y*-axis) are separated by strands and averaged between replicates. Spearman correlation coefficients are included in each panel. TCR increases with gene expression and removes damage in the TS strand; GR also increases with gene expression, but at a lower rate

**Effect of transcription on damage formation and repair**. To study the effect of transcription on cisplatin-induced DNA damage and excision repair, we used our RNA-seq data to segregate genes based on their expression quantiles in the cisplatin-treated group and investigated how DNA damage and excision repair change in the TS and NTS as a function of gene expression (Fig. 3). Our results show that, on the global scale, the damage level in the NTS was roughly uniform, and is independent of the gene expression level, due to the fact that GR in the NTS only affects a small fraction of the overall accumulated damage. For the damage level in the TS; however, we found that the remaining damage at 4 h after cisplatin injection decreases as gene expression (or equivalently, transcription-coupled repair) increases and is less than that in the NTS on the global scale (Fig. 3,

Supplementary Figure 6). The magnitude of difference between the two depends on the TCR and thus the gene expression. Furthermore, assuming that the proportion of damage repaired by GR is negligible compared to the total induced damage, we have:

$$\frac{\int \mathrm{TCR}_t \mathrm{d}t}{\sum \mathrm{Damage}} \approx 1 - \mathrm{Damage}^{\mathrm{TS}}/\mathrm{Damage}^{\mathrm{NTS}}. \qquad (3)$$

Our results indicate that liver has the highest repair efficiency by repairing the highest proportion of damage out of the total (Supplementary Figure 15) and that the overall accumulated

damage is driven by repair efficiency instead of damage formation.

Globally, we find that the excision repair measured by XR-seq in the TS increased with gene expression, but plateaued among highly expressed genes, potentially due to total repair saturation (Fig. 3). For the NTS, we found that there was less repair compared to the TS (Fig. 3, Supplementary Figure 12), and that the repair in the NTS increased at a lower rate by expression without saturation (Fig. 3), which can be explained by the open chromatin nature of the active genes.

**Effect of epigenomic profiles on damage formation and repair.**
Packaging DNA into nucleosomes or higher order chromatin structure prevents repair machinery from accessing damaged DNA, which can result in differences in DNA-damage repair kinetics[39]. We hypothesize that chromatin structure, in addition to transcriptional rate, might be the rate-limiting factor for TCR. Recent studies have shown that the efficiency of TCR of UV-induced cyclobutene (CPDs) and (6-4) pyrimidine–pyrimidone photoproducts (repaired by nucleotide excision repair) is closely associated with chromatin states[40]. In order to further investigate the effect of chromatin states, transcription factor binding, and histone modifications on the damage formation and excision repair, we retrieved the publicly available epigenomic profiles for mouse kidney, liver, lung, and spleen from the ENCODE database[17]. Supplementary Table 1 specifies the bed files that we downloaded, which contain genome-wide measurements of average intensities for each chromosomal region for H3K4me1, H3K4me3, H3K27ac, H3K27me3, H3K36me3, and POLR2A markers by ChIP-seq, as well as DNase I hypersensitivity sites by DNase-seq.

As a sanity check, we first investigated the relationship between expression and epigenomic markers in the control samples. We found that H3K27me3 is negatively correlated with the expression, while all other epigenomic profiles are positively correlated, as expected (Supplementary Figure 16). We further inferred the relationship between transcription, DNA damage and repair, and epigenomic profiles by studying the non-induced genes (i.e., genes that do not show differential gene expression after cisplatin treatment) in each organ. Our results show that excision repair in the TS is positively correlated with gene expression, H3K4me3, and DNase I markers across all organs, while the damage in the TS is negatively correlated with the aforementioned epigenetic markers (Supplementary Figure 17). We did not observe significant correlations, in the same directions across all organs, between the damage level in the NTS and the epigenetic markers. This further supports the fact that damage is relatively uniformly distributed along the genome (Supplementary Figure 17).

To compare across different organs, we first identified organ-specific gene expression using RNA-seq data for both cisplatin-treated and control samples. For simplicity and without loss of generality, we focus on comparing liver against the other organs from this point on. Specifically, we carry out pairwise differential expression analysis using DESeq2, which compares liver with each of the other organs. The intersection of the significant genes from the pairwise comparisons is shown in Supplementary Figure 18. For the genes that are significantly highly and lowly expressed in liver compared to the other organs, we further investigated their organ-specific DNA damage, excision repair, and epigenomic profiles (Fig. 4).

Specifically, for the 602 genes that are highly expressed in liver (Fig. 4a), we found that TCR is higher in liver, reflected by higher excision repair in the TS. GR is also higher in liver compared to other organs, possibly due to a greater number of active genes with open chromatin regions, as indicated by the epigenomic

markers (Fig. 4a). Notably, TCR removes DNA adducts at a much higher rate than GR, resulting in a lower amount of damage in the TS, while the overall accumulated damage in the NTS doesn't differ between organs (Fig. 4a). For the 414 genes that are lowly expressed in liver (Fig. 4b), we observed low repair in TS and NTS in liver and no significant difference in the remaining damage level across different organs (Fig. 4b). H3K4me1, H3K4me3, H3K27ac, and H3K36me3 are markers of gene activation[17,41] and are correlated with high excision repair (Fig. 4a). On the other hand, H3K27me3, a marker of inactive genes[41], is higher for genes with lower expression and excision repair (Fig. 4b). DNase I hypersensitivity, which marks the regions of open chromatin[17], is higher for both highly and weakly expressed genes (Fig. 4a, b).

So far, we have focused on gene-specific measurements of gene expression, DNA damage, excision repair, and chromatin states. We further carried out genome-wide analysis of DNA damage and repair, with association of different chromatin states. The chromatin states, shown in Fig. 4c, are derived from histone modifications and genomic sequence elements, with functional roles inferred of each state[42]. We found that, similar to previous results[18], DNA repair is higher in active chromatin states such as active promoters and CpG islands, the latter of which are enriched at transcription start sites (Fig. 4c). The distribution of the damage, on the other hand, with some minor differences, is rather uniform (Fig. 4c). The differences in damage frequency that we observe can potentially be attributed to differences in DNA repair and number of GG dinucleotides.

Finally, as an example, we focused on chromosome 11, which carries the *Per1* gene that is upregulated across all organs after cisplatin treatment (Fig. 5). As apparent from the high-resolution data that Pt-d(GpG) damage distribution on NTS was uniform among the different organs. In contrast, the repair efficiency on the TS is associated with TCR (shown as strong XR-seq signal on TS), and results in less Pt-d(GpG) damage on the TS. For repair, the region with high levels of initial repair and low levels of damage on the TS exhibited H3K4me3 or DNase I sensitivity (Fig. 5). Note that the damage level is affected by the cisplatin delivery efficiency (Supplementary Figure 7) and thus kidney has a slightly higher damage level, shown in Fig. 4. We further normalize the XR-seq and Damage-seq data by taking the ratio, TS/(TS+NTS), and obtain the same results (Supplementary Figure 19). Altogether, our data reveal that transcriptomic and epigenomic profiles are associated with excision repair and can thus recapitulate the genome-wide damage and repair profiles.

## Discussion
Understanding the details of how cisplatin and other platinum compounds activate cellular DNA damage responses and repair signaling transduction will help us develop new strategies to improve chemotherapeutic efficiency. Among the different DNA damage response pathways, nucleotide excision repair pathway constitutes the main mechanism to detect and repair cisplatin-induced DNA adducts[6]. Although the machineries and regulatory mechanisms of nucleotide excision repair are relatively well-known[7,43,44], the effect of transcriptomic and epigenomic profiles on DNA damage and repair in a tissue-specific manner is at early stage of investigation.

We have recently developed high-throughput sequencing methods to assay genome-wide DNA damage formation and its repair with single-nucleotide resolution, namely, Damage-seq and XR-seq. Damage-seq has been successfully applied to measure cisplatin- and ultraviolet-induced DNA damage in cultured human cell lines[15,18]; XR-seq has been successfully adapted to assay genome-wide excision repair in bacteria, plant, yeast, mouse, and human[18,25–27]. To our best knowledge, this is the first

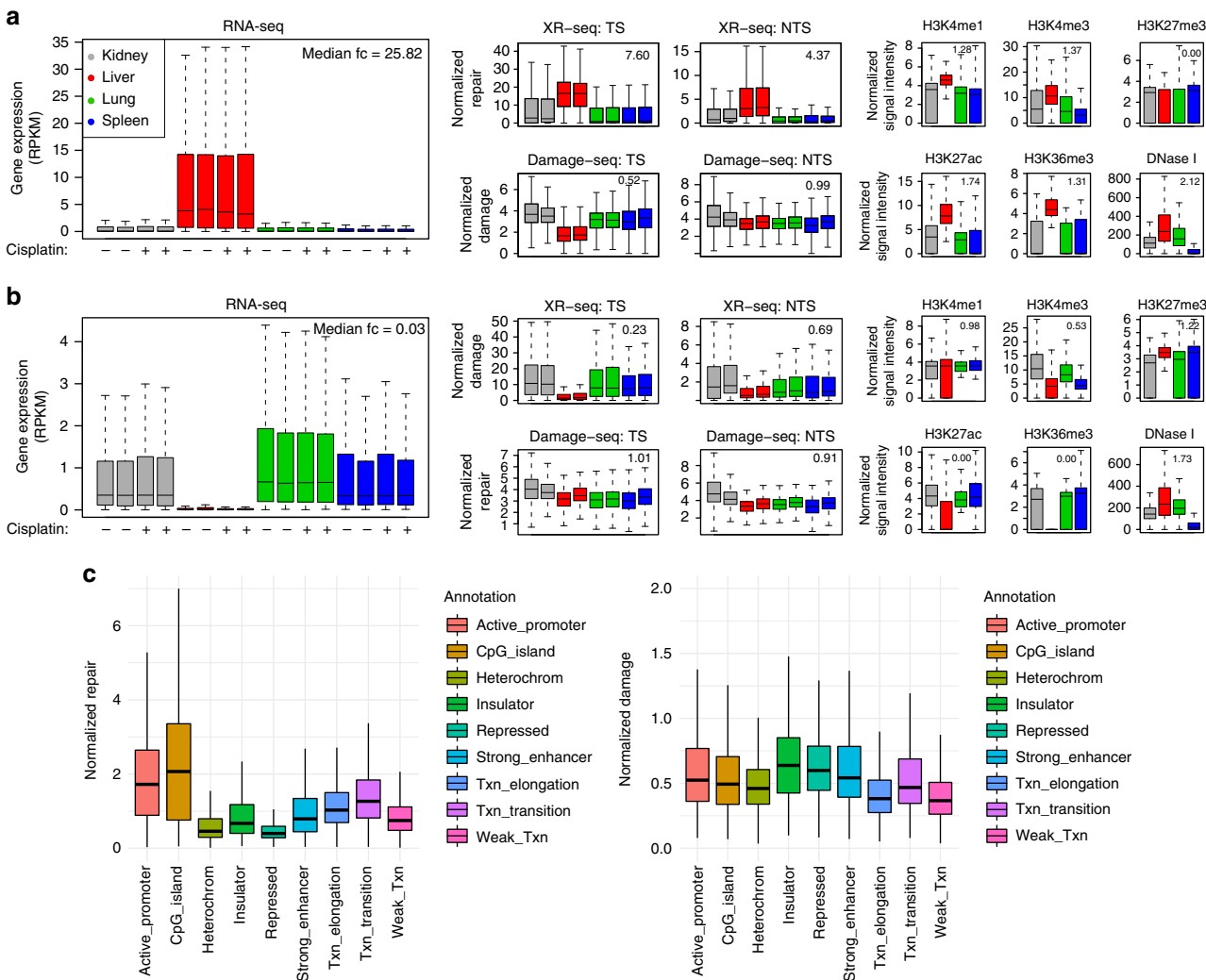

**Fig. 4** Association of DNA damage and repair with gene expression and chromatin states. **a** A total of 602 genes that are highly expressed in liver show higher transcription-coupled repair (in TS) and global repair (in NTS). Expression levels for these genes are also correlated with ChIP-seq signals for histone modification markers (H3K4me1, H3K4me3, H3K27ac, and H3K36me3). **b** A total of 414 genes that are lowly expressed in liver have lower repair and are characterized by H3K27me3, a marker for gene inactivation. Chromatin accessibility (DNase I hypersensitivity site) is higher for both groups of genes in **a** and **b**, indicating a role of chromatin states in transcriptional regulation. Median fold change (fc) between liver and the other organs across all significant genes is included on the upper right corner within each panel. **c** Whole-genome analysis results of DNA damage and repair in liver, with different genomic annotations. Analysis of repair (left) and damage (right) levels across nine genomic annotations for mouse liver reveals uniform distribution of damage but higher repair in active promotor, CpG island (enriched at transcription start sites), transcription elongation and transition regions in genome

study that quantifies cisplatin-induced DNA damage and repair across mouse organs between cases and controls on the genome-wide scale. We further combined Damage-seq and XR-seq with RNA-seq and created a comprehensive "omics" resource of DNA damage, repair, and gene expression across mouse organs. It is noteworthy that the inherent heterogeneity of tissue samples from mouse kidney, liver, lung, and spleen makes our measurements potentially biased as attenuated means across different cell types. Cell-type-specific DNA damage and repair within an organ are hard to assay. This is due to the technical and technological limitation in identifying constituent cell types in a heterogeneous tissue and gaining enough cell-type-specific excision oligos before they get degraded. As such, in this study we have resorted to "bulk" RNA-seq, XR-seq, and Damage-seq, while to develop high-throughput single-cell assay of DNA damage and repair is one of the future directions for this study and can be of great impact.

Our data show that cisplatin-induced DNA damage and repair patterns are associated with multiple factors, including transcription and chromatin states. Genome-wide Damage-seq and XR-seq data reveal that the damage formation is globally uniform and that the overall accumulated damage effect is driven by not damage formation but repair efficiency (Fig. 3). This finding is consistent with our previous high-resolution study that mapped cisplatin damage in cultured mammalian cells[18]. In addition, our data showed that the rate of excision repair on the TS and NTS of active genes is positively correlated with gene expression—the repair in the TS and NTS increases with gene expression and plateaus in the TS among the highly expressed genes (Fig. 3). Our data also suggest that cellular transcription process also stimulates repair of damage in the NTS due to the fact that transcription is associated with open chromatin conformation and hence increased accessibility to the repair machinery (Figs. 3 and 4). These findings are consistent with earlier results obtained using a primer extension technique in *S. cerevisiae*[45], and results from high-resolution repair maps of UV-induced DNA damages (6-4-photoproducts and CPD) in cultured human cells[16].

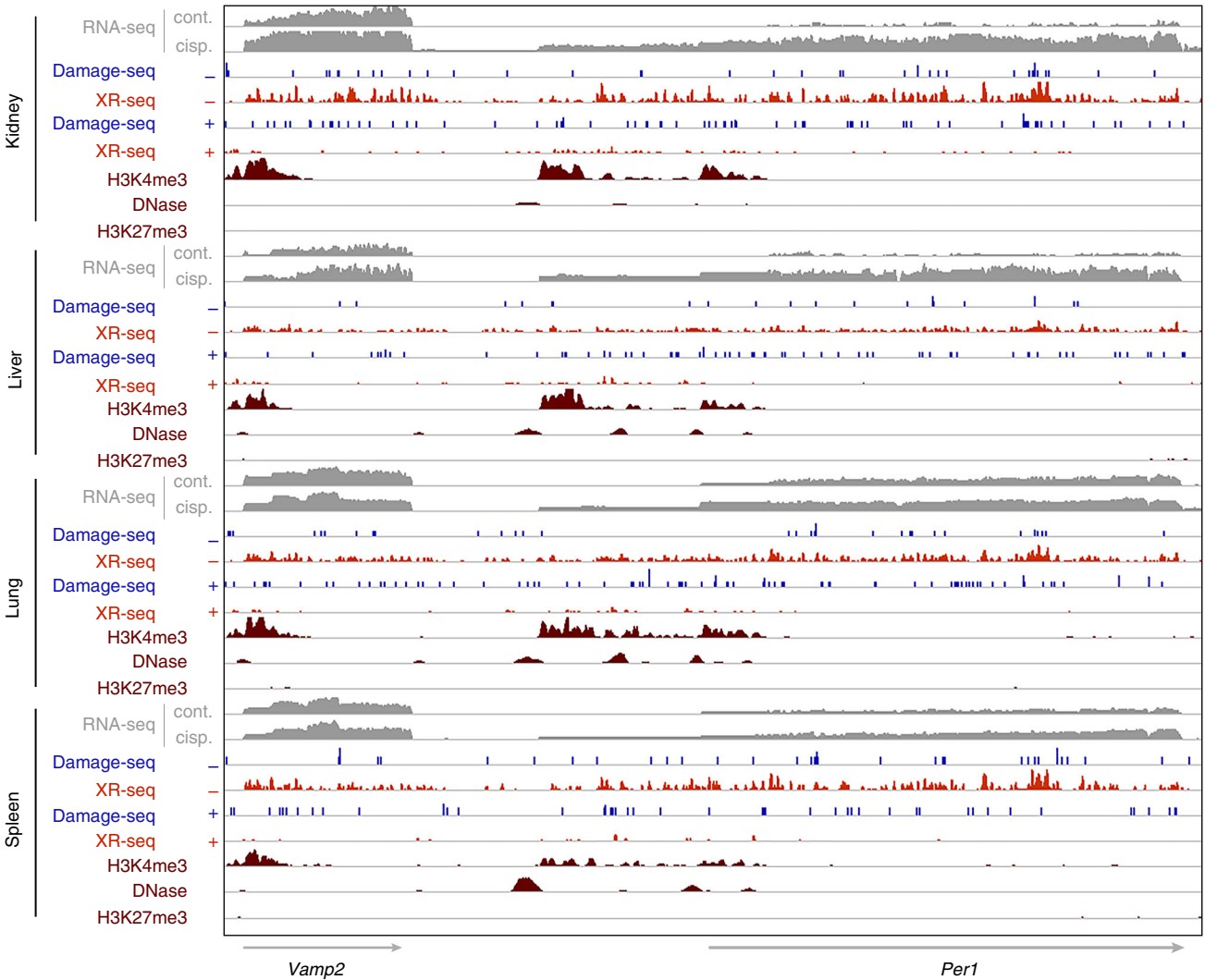

**Fig. 5** DNA damage, repair, gene expression, and epigenomic markers for *Per1*. *Per1* is significantly upregulated after cisplatin treatment across all four organs. Damage-seq and XR-seq data are shown for both strands. Pt-d(GpG) damage (Damage-seq) and repair (XR-seq) distribution on the TS and NTS are shown with − and +, respectively. Epigenetic data from ChIP-seq of H3K4me3 and H3K27me3, as well as DNase-seq, are plotted at the bottom of each organ section. We show that the transcriptional and epigenomic profiles of *Per1* and neighboring region across all four organs recapitulate the differences in DNA damage and repair between the TS and NTS

It is well documented that the acquisition of cisplatin resistance in cancer cells is a complicated process, where increased damage repair efficiency, reduced drug uptake, and increased drug efflux may potentially play a role. Our results show that spleen has the lowest amount of accumulated DNA damage induced by cisplatin (Supplementary Figure 7). GO enrichment analysis indicates that most of the downregulated genes in spleen belong to cell membrane organization, membrane invagination, cellular response to metal ion, and ferric iron transport by GO annotations. Previous studies have shown that deletion of *Ctr1* (copper transporter) in both yeast and mouse cells causes a decrease in intracellular cisplatin levels, suggesting a connection between cellular copper and cisplatin transport[46]. We found that *Atp7b* and *Steap3*, two genes that play a role in iron and copper homeostasis, were downregulated in response to cisplatin treatment in spleen. ATP7B is a copper-transporting P-type ATPase, which is associated with cisplatin transport across the plasma membrane and across the cell[47]. Overexpression of ATP7B is associated with resistance of various cancer cells to platinum drugs by increasing drug efflux[1,48]. We found that *Atp7b* expression is decreased

(four fold) in spleen after being treated with cisplatin, although the exact mechanism, by which *Atp7b* is downregulated in spleen, is not clear.

In addition to nuclear DNA damage, the cellular mitochondrial DNA is also a target of cisplatin, suggesting that mitochondrial DNA damage may contribute towards cisplatin-induced cell death. Furthermore, it has been reported that the cisplatin analogue oxaliplatin relies on a different mechanism—ribosome biogenesis stress—to kill cancer cells, suggesting that a different mechanism exists in addition to DNA-damage response to induce cell death between platinum-like chemotherapeutic drugs[49]. Given this aspect, studying the mechanisms behind the coordination of nuclear and mitochondrial DNA damage and repair may be of future interest by means of improving drug efficacy while reducing side effects to normal tissues.

Here, we present a comprehensive genome-wide study of DNA damage and excision repair across four mouse organs and investigate their associations with transcription and chromatin states, which serves as resource for researchers interested in DNA damage and repair in mouse models. This framework can be

readily applied to other studies with a focus on chemical perturbations, drug resistance, and optimization of cancer chemotherapy regimens.

## Methods

**Animal experiments**. Six or seven-month-old wild-type (C57BL/6J) female mice weighing 25–28 g were purchased from the Jackson Laboratory (Bar Harbor, ME). Without further indications, mice were maintained under a 12-h light/12-h dark regimen. All animal care and handling were performed according to the NIH and the University of North Carolina School of Medicine (Institutional Animal Care and Use Committee) rules for animal protection. Unless otherwise noted, all mice were treated with drug at ZT1 (8 am) and killed at ZT5 (12 pm). A single non-lethal dose of clinical cisplatin (1 mg/ml or 3.3 mM/L, by Fresenius Kabi) was administered by intraperitoneal injection at 10 mg cis-DDP/Kg body weight. We chose 10 mg/Kg dose to ensure sufficient Pt-DNA adducts formation for detecting genome-wide damage and repair by Damage-seq and XR-seq methods. This dose has been routinely used in studies with the mouse strain C57BL/6J and is shown to be well tolerated by the strain[50]. We followed this common practice and in agreement with previous studies, we observed no morbidity or mortality for the duration of our experiment and for at least 10 days following the administration of the cisplatin. An identical volume of saline solution (154 mM) was administered by intraperitoneal injection to control group mice.

**Assay of gene expression by RNA-seq**. Total RNA was extracted from mouse kidney, liver, lung, and spleen using TRIzol RNA extraction (Thermo Fisher Scientific). After phase separation, RNA was purified using the PureLink RNA Mini Kit (Thermo Fisher Scientific). Purified total RNA sample was treated with DNase I (TURBO DNA-free™ kit, Ambion Inc.) to remove traces of DNA. RNA libraries were prepared using the NEBNext Ultra RNA library prep kit (Illumina) according to the manufacturer's instructions. In brief, total RNA starting with 1 μg was poly-A selected, fragmented by random priming and then converted to cDNA using ProtoScript II reverse transcriptase. The cDNA was then end-repaired, adenylated and ligated with Illumina sequencing adaptors. Libraries from all samples were pooled and sequenced using an Illumina HiSeq 4000 with 50-bp pair-end read.

**Assay of DNA damage by Damage-seq and immuno-slot blot**. In Damage-seq assay[15], genomic DNA was extracted using PurLink Genomic DNA kit (Thermo). Ultrasonic fragmented genomic DNAs were purified using an equal volume of HighPrep PCR beads (MagBio). Purified DNA (~1 μg) was used for End-repair and dA-tailing and adaptor ligation (NEBNext Ultra II DNA Library Prep Kit) following manufacturer's instructions. Following IP (immunoprecipitation) with 0.4 μg anti-cisplatin antibody (ab 103261, Abcam), the DNA was primer extended in the presence of 30 pmol Bio3U (biotin elongation primer, 5′-bio-AGAGTG/dU/GACTGGAGTTCAGACGTGTGCTCTTCCGATCT-3′) using NEBNext Q5 Hot Start HiFi PCR Master Mix. After undamaged DNA strands were captured by 20 pmol (2 μL) of SH oligo (SH, Subtractive hybridization, 5′-bio-NNGACTGGTTCCAATTGAAAGTGCTCTTCCG-SpC3-3′), DNAs were purified using phenol-chloroform extraction and ethanol precipitation. The DNA was then ligated to a second adaptor ligation using T4 DNA ligase HC (Thermo) at 16 °C overnight. After quality check, the DNA samples were purified with HighPrep PCR beads, and ligated DNAs were PCR amplified by NEBNext Ultra II PCR Master Mix with NEBNext Multiplex Oligos for Illumina (New England Biolabs). The PCR libraries were purified with HighPrep PCR beads. Libraries from all samples were pooled and sequenced using an Illumina HiSeq 4000 with 50-bp single-end read.

For the immuno-slot blot assay, a total of 200 ng genomic DNA was spotted onto a PVDF membrane (Immun-Blot LF, Bio-Rad Laboratories Inc.) using a transfer device (Bio-Dot SF, 160BR 08600, Bio-Rad Laboratories Inc.). After blocking in 5% skim milk, the membrane was incubated with cisplatin antibody (1:2000 dilution in 5% milk) overnight at 4 °C. The membrane was washed three times with 1X TBST (0.2% Tween 20), then incubated with secondary (1:5000, anti-rat HRP, GE Health Care, NA 935V) antibody for 1 h at room temperature. After washing the membrane, signals were detected with Clarity Western ECL chemiluminescent reagent (Bio-Rad Laboratories Inc.). In order to visualize the total DNA signal from each individual spot as an internal control, the membrane was washed three times with 1X TBST, and incubated with anti-DNA antibody (1:10,000, anti-DNA antibody, MAB 3034, Millipore Sigma) overnight at 4 °C. The membrane was washed three times, then incubated with secondary antibody (1:5000, anti-mouse HRP, GE Health Care, NA 931V) for 1 h at room temperature, then the same washing and signal detection steps were repeated.

**Assay of DNA excision repair by XR-seq**. In XR-seq assay[18], 4 h after treatment of mice with cisplatin, the mice were killed by carbon dioxide exposure, the kidneys, liver, lung, and spleen were removed and washed extensively with cold phosphate buffered saline (PBS), and then homogenized in 5 ml ice-cold PBS until the plunger moved freely (around 15–20 strokes based on the source of tissue) using Telfon homogenizer. The homogenized tissues were transferred into 50 ml tubes and washed three times with ice-cold PBS at 2500 rpm for 4 min. After last wash, the cell pellets were resuspended in 5 ml ice-cold buffer A (25 mM HEPES,

pH 7.9, 100 mM KCl, 12 mM MgCl₂, 0.5 mM EDTA, 2 mM DTT, 12.5% glycerol and 0.5% NP-40) and incubate on ice for 10 min. Then, cells were lysed by an ice-cold Dounce homogenizer with 70 strokes using tight plunger. The chromatin fraction was then pelleted by centrifugation for 30 min at 14,000 rpm at 4 °C in a 1.7 ml microcentrifuge tubes. The pelleted chromatin fragment was kept at −20 °C for further DNA extraction and damage-seq library preparation. The supernatants containing low molecular DNA-protein fragments (excision products) were immunoprecipitated with anti-TFIIH (p89 antibody (G-10, sc-271500, and p62 antibody (H-10, sc-25329), Santa-Cruz Biotechnology), and ligated to adaptors on the both ends. After a second immunoprecipitation with 2.5 μg cisplatin-specific antibody, the Pt-DNA adducts were reversed by incubating in NaCN (200 mM) overnight at 65 °C. Then the damage-free DNAs were amplified by PCR to get XR-seq libraries and sequenced using an Illumina HiSeq 4000 with 50-bp single-end read.

**ENCODE data**. Epigenomic data across four mouse organs were downloaded from the ENCODE portal as bed files and visualized using the UCSC browser (Supplementary Table 1). All of the ENCODE data were generated from mice without cisplatin treatment, which enabled us to study DNA damage and repair within the context of pre-existing epigenomic profiles. While the effect of cisplatin treatment on epigenomic profiles would be of interest, here we focused on the genes that do not show significant up- or down-regulation after cisplatin treatment across all organs tested. In addition, most of the ENCODE data were generated using 8-week-old young adult mice (Supplementary Table 1).

**Bioinformatic processing for Damage-seq, XR-seq, and RNA-seq**. For Damage-seq, cutadapt[51] was used to remove reads adaptor sequence GACTGGTTCCAATTGAAAGTGCTCTTCCGATCT at the 5′-end, which are from the undamaged strands[15]. For XR-seq, cutadapt[51] was used to trim reads with adaptor sequence TGGAATTCTCGGGTGCCAAGGAACTCCAGTNNNNNNACGATCTCGTATGCCGTCTTCTGCTTG at the 3′-end and discard untrimmed reads[16]. For both Damage-seq and XR-seq, BWA[52] was used for read alignment, and Picard tools (http://broadinstitute.github.io/picard/) was used for filtering, sorting, deduplication, and indexing. Post-alignment filtering steps were adopted using Rsamtools (http://bioconductor.org/packages/Rsamtools/) for XR-seq and Damage-seq, respectively. For XR-seq, we only keep reads that (i) have mapping quality >20, (ii) are of length 21–31 bp, and (iii) have Pt-d(GpG) dinucleotide sequence 5–8 bp upstream from the 3′-end of the reads. For Damage-seq, we only kept reads that (i) had mapping quality >20, (ii) had Pt-d(GpG) dinucleotide sequence 1–2 bp upstream from the 5′-end of the reads. Reads from the TS and NTS strands were separated using known gene annotations for mm10 by ENSEMBL.

For RNA-seq, reads were aligned using STAR[53], followed by a filtering step to remove: (i) reads with mapping quality less than 20, (ii) read pairs with unexpected orientations and >500 Kb mapping distance between the read pair, and (iii) reads that were mapped to >10 positions in the genome. Read counts for each gene were obtained using FeatureCounts[54], followed by DESeq2[31] for differential expression analysis, with a threshold of 0.4 on log fold change and a false discovery rate cutoff of 0.05. Gene ontology analysis was performed using http://geneontology.org, with a false discovery rate cutoff of 0.05. GO enrichment results were further processed with REViGO[55] and visualized by Cytoscape[56], as shown in Supplementary Figure 14.

**Quality control and data normalization**. We use RPKM for within-sample normalization for the RNA-seq data and adjust for sample-specific sequencing depth and gene-specific number of GG dinucleotides for XR-seq and Damage-seq. For DESeq2 differential expression analysis, we use all genes as input to the software. To investigate the relationship between gene expression, chromatin states and DNA damage and repair, we adopted a stringent quality control (QC) procedure and only retained genes that: (i) had at least one sample with expression; (ii) had matched gene ID across all sequencing platforms; (iii) had at least ten GG dinucleotides in the TS or the NTS; (iv) were less than 100 Kb; and (v) have at least 20 reads in total across samples for RNA-seq, Damage-seq, and XR-seq, respectively. Batch effect was adjusted using ComBat[30] with batch as a known covariate.

For data normalization, we also took the ratio of the reads from the TS and the NTS, TS/(TS+NTS), to remove biases and artifacts that are shared between the two DNA stands (i.e., library size, number of di-guanines/gene length, cisplatin delivery efficiency (Supplementary Figure 7)), as well as other gene-specific biases, such as sequencing bias and antibody pull-down efficiency, etc. The ratio is bound between 0 and 1 and sheds light upon how TCR and GR interplay.

**Reporting Summary**. Further information on experimental design is available in the Nature Research Reporting Summary linked to this Article.

## Data availability

The data reported in this paper have been deposited in the Gene Expression Omnibus (GEO) database, www.ncbi.nlm.nih.gov/geo (accession number GSE117167). All code used in this paper is available at https://github.com/

yuchaojiang/damage_repair. All other relevant data are available from the corresponding authors on request.

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

## Acknowledgements

This work was supported by NIH Grants GM118102 (to A.S.), ES027255 (to A.S.), CA142538 (to Y.J.), and a developmental award from the UNC Lineberger Comprehensive Cancer Center 2017T109 (to Y.J.), and an award from Scientific and Technological Research Council of Turkey 118C023 (to O.A.). We thank Dr. Jinchuan Hu and Dr. Christopher Selby for helpful discussions and comments. We also thank Ayano Kakoki for help with the illustration of mouse, syringe, and mouse organs in Fig. 1a.

## Author contributions

A.S. envisioned and initiated the study, while A.Y. performed the experiment. A.Y., O.A., A.S. and Y.J. planned and executed the data analysis. A.Y. and Y.J. wrote the manuscript, which was further edited and approved by all authors.

## Additional information

**Competing interests:** The authors declare no competing interests.

