## [Peer Review File · Nature Communications]

Reviewers' comments:

Reviewer #1 (Remarks to the Author):

The manuscript by Yimit et al describes an analysis of cisplatin induced DNA-adducts across four mouse organs four hours following intraperitoneal administration of cisplatin. The data show the potential to use genome-wide approaches to analyse cisplatin damage formation and excision repair in vivo following platinum treatment. Different DNA damage and repair spectra are observed, although it is difficult to make clear conclusions regarding cellular responses to cisplatin of different tissues and how this may affect dose limiting toxicities (DLTs).

1. Kidney, liver, lung and spleen tissues are examined. Renal toxicity is one of the main DLTs for cisplatin, although myelosuppression is also important. Haematological cells have not been examined in the present study, other than spleen.
2. The dose of cisplatin used is presumably highly toxic in this strain of mice. The maximum tolerated dose of cisplatin is around 6mg/kg in BALB/C mice, so I assume 10mg/kg will be lethal in the strain used. This needs to be commented on.
3. Cisplatin takes about 12 h to maximally form crosslinks. Are monofunctional adducts mainly being measured at 4 hours? The enrichment of GpG dinucleotides is encouraging. Is there enrichment for other dinucleotides (ApG, CpG, etc), and lack of enrichment for non-relevant ones?
4. An analysis of genomic regions cisplatin adducts are forming at would be important beyond gene specific regions e.g. repetitive sequences, CpG islands, etc.
5. Aspects of the interpretation of the data are highly speculative. For instance, while the tissue specific gene expression data is useful to present, it adds little to the study and demonstrating a role for Per1 and circadian rhythms would need further work. I would recommend this not being included in the abstract and the discussion in results and discussion section shortened.
6. The work presented is largely descriptive. Elegant and detailed bioinformatic analysis has been done, however little statistical interpretation of the data presented is given to support the associations observed.
7. The data presented shows the feasibility of analysing genome wide cisplatin adducts and repair and provides a potential framework for future studies. However, the data do not provide conclusions concerning why renal toxicity can be dose limiting for cisplatin and how this information can be used to tailor cancer chemotherapy regimens.

Reviewer #2 (Remarks to the Author):

This manuscript by Yimit et al. catalogs and quantifies sites of DNA damage and repair and integrates them with RNA-seq data along with community-derived epigenetic data for four mouse tissue. This is an ambitious study whose results might be useful to study DNA-damaging drugs such as cisplatin. While this is likely the first study that quantifies cisplatin-induced DNA damage across mouse organs on a genome-wide scale, it is largely difficult to read. I believe that the manuscript is publishable in Nature Communications after the following issues to improve the manuscript are incorporated:

1. Figure 1 – In panel a, under “No treatment,” it is indicated as “epigenetics (ENCODE)” – this should be changed as the term epigenetics is not informative enough and especially in the context of this experimental strategy. The corresponding text in lines 70-71 should also be expanded to discuss specifically what data “epigenetic profiles” refers to.
2. The authors took methylation, acetylation and DNase hypersensitivity site data from the ENCODE project. Perhaps they could also discuss some caveats of doing this as opposed to obtaining such data from the same mice in which the cisplatin experiments were done. Also, was the age of mice used in the authors’ experiments the same as those used in the ENCODE project?

3. The titles of sections "Tissue-specific cisplatin DNA damage maps" and "Tissue-specific cisplatin DNA repair maps" are somewhat misleading. One expects to read about certain changes that are specific to some tissues but that is not discussed. It is unclear what "tissue-specific" really refers to. Also, if only four tissues are sampled, no claim of any type of "tissue-specificity" can be made anyway. Also, the statement in lines 147-149 that "Our Damage-seq and XR-seq data show organ-specific DNA damage and repair patterns (Fig.1b), suggesting that there is a mechanistic link between cisplatin induced DNA damage, repair, and gene transcription regulation" does not logically follow from the data, which are simply PCA plots.

4. Line 43: "...were generated using isolated cell lines (including tumors)" – it is not clear what "including tumors" refers to when they are mentioning cell lines.

5. The authors state "We didn't find enrichment terms for circadian rhythm in liver and spleen. I think the authors are confused between genes involved in circadian rhythm and whether they are involved with "statistical enrichment" of the class of genes annotated with "circadian rhythm" as GO terms.

6. The section on integration of the data with epigenetic markers (H3K4me1, H3K4me3, H3K27ac, 235 K3K27me3, H3K36me3, POLR2A, and DNase I markers) is underwhelming. This section could be expanded upon.

7. There are a number of grammatical errors throughout the manuscript. A few are highlighted below:

i) Line 36. Change "Reoccurrence" to "recurrence"

ii) Line 50: "...when extending the application to the in vivo situation." Not clear what "situation refers to.

iii) Line 63-64: "Specifically, cisplatin was administered by intraperitoneal injection in mouse."

iv) Lines 175-176: "...circadian rhythm and rhythmic process were enrichment among the genes..."

v) Lines 178-179: "We didn't find enrichment terms for circadian rhythm in liver and spleen." I believe that the authors are a bit confused between involvement of certain genes

vi) Line 247-249: "For simplicity and without loss of generality, we focus on comparing liver against the other organs from this point on, while the results generalize." This statement should be rewritten.

vii) "...we further investigate how their excision repair and epigenomic profiles, with results shown in Fig 4."

Reviewer #3 (Remarks to the Author):

The manuscript by Yimit et al generates damage and repair maps after 4 hours of cisplatin treatment in 4 mouse organs. Other cisplatin damage and repair maps have been published by the same lab although in cell lines (see Hu et al., PNAS 2016). The novelty of this paper relies in the in vivo characterization of the damage and repair and comparison between 4 different organs.

Data is interesting and valuable, although some of the major conclusions obtained are confirmation of previous work on cell lines.

Some questions and comments for the authors:

The protocol involves "Total RNA was extracted from mouse kidney, liver, lung and spleen" 4h after cisplatin treatment. However different organs have a different mixture of cell types that may have very different expression and epigenomic profiles, they may also have different tolerance to damage and different level of excision repair. While liver is quite homogeneous in terms of cell types within the organ, kidney is a very complex organ with many different cell types. Conclusions of the amount of damage and repair at the level of organ may mask differences within cell types. For example, a medium amount of excision repair, may be due to some cell types with very high level of repair and others cell types in the same organ with very low repair.

How do authors deal with this? Some comments on the paper about this complexity would be useful.

Is the protocol of ENCODE to generate the epigenome data of the mouse comparable in terms on how and when RNA was extracted? Do they also take RNA from the whole organ? Are mice of similar age? If this were not the case, it does not invalidate the analysis, but would be interesting to inform in the paper about possible differences in the protocol if so, and what can they mean when interpreting the results.

Figure 1. Why XR-seq has not been done in control group?

It seems, although it is not completely clear from the text, that all measurements are at the level of genes. Could authors clarify this?

For example, in supplementary figure 15 and 16, each dot in the correlation plots is a gene? It is necessary that this is clarified in methods and figure legend.

The damage and repair data are not restricted to genic regions, neither it is the epigenome data from ENCODE. Why the authors have restricted their analysis to genic regions? Couldn't they have analyzed the rest of the genome to make a more complete study?

Page 6. "We then applied PCA to the normalized read count matrix and found that samples from the same organ are clustered together, indicating organ-specific DNA damage (Fig. 1b)"

That could be due to organ specific DNA repair during the 4 hours.

Any idea, hypothesis on why spleen is a minor site for cisplatin damage, while kidney is a major site? Is this related to accessibility of the drug? Its metabolism?

Is there any correlation between repair levels in different organs and expression of NER pathway genes?

Supplementary figure 7, in spleen the mitochondrial normalized damage is equivalent to the non-mitochondrial, why is that?

"5.4% of reads mapping to mitochondrial DNA contain GG, which indicates that there is no excision repair". What the other reads mean in an XR-seq experiment? Noise?

Seems that expression analysis also shows a lower impact of cisplatin in spleen compared to other tissues (i.e. only 193 genes differentially expressed), consistent with lower amount of damage. Why is that? Accessibility of the drug to the organ?

The authors say "We did not observe significant correlations between the damage level in the NTS and the epigenetic markers, indicating a relatively uniform damage distribution along the genome". But looking at supplementary figure 16 one of the strongest correlations (anticorrelation in this case) is between Damage (TS and NTS) and H3K36me (-0.46 and -0.46 in kidney). How do authors explain this?

Figure 4. How many genes are considered as "highly expressed in liver" in the analysis of Figure 4a, and how many as "weakly expressed in liver" in Figure 4b? This should be clarified.

We thank the editor and reviewers for their careful reading, encouraging comments, and constructive suggestions. Below are our point-by-point responses to the reviewers' comments. The original reviewers' comments are in *italics* and our responses are in normal blue font. The changes that we made in the manuscript are highlighted in yellow.

Reviewer #1:

The manuscript by Yimit et al describes an analysis of cisplatin induced DNA-adducts across four mouse organs four hours following intraperitoneal administration of cisplatin. The data show the potential to use genome-wide approaches to analyze cisplatin damage formation and excision repair in vivo following platinum treatment. Different DNA damage and repair spectra are observed, although it is difficult to make clear conclusions regarding cellular responses to cisplatin of different tissues and how this may affect dose limiting toxicities (DLTs).

1. Kidney, liver, lung and spleen tissues are examined. Renal toxicity is one of the main DLTs for cisplatin, although myelosuppression is also important. Haematological cells have not been examined in the present study, other than spleen.

We agree with the reviewer that myelosuppression is an important toxicity along with nephrotoxicity. Cisplatin related myelosuppression has been examined previously in many studies. In our study, we mostly focus on cisplatin-induced DNA damage and repair across mouse organs. It has been shown that DNA adducts in blood samples from cisplatin-treated rat (after one-hour treatment with 10mg/Kg of body weight) are too low to be measured. The main Pt-d(GpG) adducts level is lower than spleen (Fichtinger-Schepman, et al., 1989).

Similarly, in C57Bl/6 mouse about six- and three-times lower cisplatin-induced DNA adducts was observed in peripheral T lymphocytes than kidney and liver cells after exposed to 10mg/Kg of body weight cisplatin (Liedert, et al., 2006). We believe that as a body's largest filter and reservoir of blood, spleen-based transcriptome, DNA damage and repair analyses partially represent haematological cells study.

2. The dose of cisplatin used is presumably highly toxic in this strain of mice. The maximum tolerated dose of cisplatin is around 6mg/kg in BALB/C mice, so I assume 10mg/kg will be lethal in the strain used. This needs to be commented on.

We administered cisplatin once at a dose of 10mg/Kg, as outlined in the "Recommended Administration Values by Route" for rodents according to rules for animal protection by the NIH and the University of North Carolina School of Medicine (Institutional Animal Care and Use Committee). While we chose 10mg/Kg to ensure sufficient DNA-platination and excision oligos across various mouse tissues for Damage-seq and XR-seq measurements, this dose that we adopted is not lethal for the mouse strain that we used. After careful literature search, we further found that various maximum tolerated doses exist between mouse strains. Specifically, cisplatin used at 6 mg\Kg is only toxic to BALB/c mice (20% weight loss) in the second dosing cycle. It has also been shown that only 1/3 of C57BL/6J mice lost more than 20% weight 4 days after 8 mg/Kg cisplatin was used (Aston, et al., 2017), which indicates that the C57BL/6J mice could tolerate higher cisplatin dose than BALB/c mice.

3. Cisplatin takes about 12 h to maximally form crosslinks. Are monofunctional adducts mainly being measured at 4 hours?

Previous studies (Dakup, et al., 2018; Liedert, et al., 2006) show that cisplatin makes maximum organ-specific DNA crosslinks within 12 hours. For our experiment, as outlined in Figure 1, we measured cisplatin-induced DNA damage, repair, and gene expression 4 hours after cisplatin was administered, and we found significant GpG dinucleotide enrichment in both Damage-seq and XR-seq (Supplementary Figure S2 and S10). More importantly, we found that the damage introduced by cisplatin treatment has not been fully repaired at 4 hours and yet the

transcription-coupled repair and global repair still occurs at a high rate (Figure 3). We therefore believe that the 4 hours' time point is a good choice to simultaneously measure DNA damage and repair.

The enrichment of GpG dinucleotides is encouraging. Is there enrichment for other dinucleotides (ApG, CpG, etc), and lack of enrichment for non-relevant ones?

Thanks! We have carried out a complete dinucleotide enrichment analysis using data from Damage-seq. Results across all organs are included in the manuscript as Supplementary Figure 3; result for the kidney sample is shown below. While we do not observe other dinucleotide enrichment, we observe a preference for A (adenine) 5' to the GG dinucleotides of our Damage-seq reads, which has been previously reported (Hu, et al., 2016).

We also included the following comment in the Results section: “Notably, while we do not observe other dinucleotide enrichment, there is a preference for adenine 5' to the GG dinucleotides of our Damage-seq reads (Supplementary Figure S3).”

4. An analysis of genomic regions cisplatin adducts are forming at would be important beyond gene specific regions e.g. repetitive sequences, CpG islands, etc.

Thanks for pointing this out. We have carried out whole-genome analysis of DNA damage and repair, with genomic annotations. Results are included in the newly generated Figure 4c and are highlighted in the Results section under “Effect of chromatin states and epigenetic profiles on damage formation and repair.” Briefly, we have observed higher repair in active chromatin states, such as active promoters and CpG islands (enriched at transcription start sites). For repetitive sequence, there is no such state previously reported for mice from histone modifications and genomic sequence elements (Bogu, et al., 2015). More importantly, with the short excised oligos and read lengths of XR-seq (peaked at 26 nt—see Supplementary Figure S9), it is hard to uniquely map reads to repetitive regions with low mappability.

5. Aspects of the interpretation of the data are highly speculative. For instance, while the tissue specific gene expression data is useful to present, it adds little to the study and demonstrating a role for Per1 and circadian rhythms would need further work. I would recommend this not being included in the abstract and the discussion in results and discussion section shortened.

We excluded Per1-related contents from abstract, shortened our discussion on the potential significance of Per1 upregulation with regard to circadian rhythm, and emphasized that “This study provides a platform for further research on optimizing cisplatin treatment efficacy and reducing side effects.”

6. The work presented is largely descriptive. Elegant and detailed bioinformatic analysis has been done, however little statistical interpretation of the data presented is given to support the associations observed.

Thanks for your encouraging comments and valuable suggestions! In this revision, we emphasized that the framework and data we presented provide a platform and serve as a resource for future research. As such, a large part of this study has been descriptive. Nevertheless, we have, in this revision, carried out detailed statistical analysis to better support the observed associations. Specifically, in this round of revision, we have included in Figure 3 the Spearman correlation coefficients to investigate the effect of transcription on DNA damage and repair. We have also included in Figure 4, for all panels of comparisons, median fold changes between liver and the other organs across all significant genes.

7. The data presented shows the feasibility of analyzing genome wide cisplatin adducts and repair and provides a potential framework for future studies. However, the data do not provide conclusions concerning why renal toxicity can be dose limiting for cisplatin and how this information can be used to tailor cancer chemotherapy regimens.

As the reviewer points out, based on our cisplatin-induced DNA damage and repair data, it is difficult to make clear conclusions regarding renal toxicity being a dose limiting factor for cisplatin treatment. Nevertheless, we believe that this information would be of value in future studies on the subject, and we hope this information will be of use in tailoring other chemotherapy regimens that include cisplatin.

Reviewer #2:

This manuscript by Yimit et al. catalogs and quantifies sites of DNA damage and repair and integrates them with RNA-seq data along with community-derived epigenetic data for four mouse tissue. This is an ambitious study whose results might be useful to study DNA-damaging drugs such as cisplatin. While this is likely the first study that quantifies cisplatin-induced DNA damage across mouse organs on a genome-wide scale, it is largely difficult to read. I believe that the manuscript is publishable in Nature Communications after the following issues to improve the manuscript are incorporated:

1. Figure 1 – In panel a, under “No treatment,” it is indicated as “epigenetics (ENCODE)” – this should be changed as the term epigenetics is not informative enough and especially in the context of this experimental strategy. The corresponding text in lines 70-71 should also be expanded to discuss specifically what data “epigenetic profiles” refers to.

For better clarity of our experimental design, we substituted “No treatment” with “ENCODE” in Figure 1 for the epigenomic data. The corresponding text was expanded accordingly as follows: “In addition, we retrieved publicly available epigenomic profiles across the four organs by ChIP-seq and DNase-seq (H3K4me1, H3K4me3, H3K27ac, H3K27me3, H3K36me3, POLR2A, and DNase I) from the ENCODE database. All of the ENCODE epigenetic data were generated from mice without cisplatin treatment, and were only investigated for the genes that do not show significant up- or down-regulation after cisplatin treatment.”

2. The authors took methylation, acetylation and DNase hypersensitivity site data from the ENCODE project. Perhaps they could also discuss some caveats of doing this as opposed to obtaining such data from the same mice in which the cisplatin experiments were done. Also, was the age of mice used in the authors’ experiments the same as those used in the ENCODE project?

Thanks! We wished to put our damage and repair data within the context of pre-existing epigenomic profiles. We do agree with the reviewer that the effect of cisplatin on the epigenomic profiles would be of interest. However, such an expansive study is outside the scope of this work, which provides a platform for future studies. In this revision, we generated a new table as Supplementary Table S4, which includes the accession number as well as the age of mice for the ENCODE data that we utilized. As pointed out, we used 6- or 7-month old wild type adult mice (C57BL/6J). Meanwhile, most of the ENCODE data we downloaded were generated using 8-week-old young adult mice.

Supplementary Table S4. Publicly available epigenomic data from ENCODE. Epigenomic data for mouse organs were downloaded from the ENCODE portal (<https://www.encodeproject.org>) as bed files and viewed using the UCSC browser and the Integrative Genomics Viewer. Data accession number as well as age of mice are included for each marker across all organs.

Organ	H3K4me1	H3K4me3	H3K27ac	H3K27me3	H3K36me3	POLR2A	DNase I
Kidney	ENCFF764QKN (8 weeks)	ENCFF039JTO (8 weeks)	ENCFF660CQT (8 weeks)	ENCFF710FSC (8 weeks)	ENCFF996FJV (8 weeks)	ENCFF160ZYB (8 weeks)	ENCFF260THC (postnatal)
Liver	ENCFF189WSW (8 weeks)	ENCFF625DES (8 weeks)	ENCFF776JLM (8 weeks)	ENCFF429PBK (8 weeks)	ENCFF632JAR (8 weeks)	ENCFF125NGA (8 weeks)	ENCFF235KRN (postnatal)
Lung	ENCFF582IWI (8 weeks)	ENCFF543QNI (8 weeks)	ENCFF388BOS (postnatal)	ENCFF873ZSM (postnatal)	ENCFF176JQW (postnatal)	ENCFF460DBD (8 weeks)	ENCFF251HWU (postnatal)
Spleen	ENCFF817MPF (8 weeks)	ENCFF574OJR (8 weeks)	ENCFF092BMP (8 weeks)	ENCFF970VMF (8 weeks)	ENCFF362AJY (8 weeks)	ENCFF518LVB (8 weeks)	ENCFF553EPA (8 weeks)

3. The titles of sections “Tissue-specific cisplatin DNA damage maps” and “Tissue-specific cisplatin DNA repair maps” are somewhat misleading. One expects to read about certain changes that are specific to some tissues but that is not discussed. It is unclear what “tissue-specific” really refers to. Also, if only four tissues are sampled, no claim of any type of “tissue-specificity” can be made anyway.

Thanks for pointing this out. While we agree that tissue-specific changes are not the main focus on this study, in Figure 4, we investigated the liver-specific gene expression, DNA damage, and repair, and show how it correlates with the epigenetic profiles. Nevertheless, to avoid potential confusion, we removed “tissue-specific” in the title of the manuscript and changed “tissue-specific” to “across organs” in the corresponding section titles.

Also, the statement in lines 147-149 that “Our Damage-seq and XR-seq data show organ-specific DNA damage and repair patterns (Fig.1b), suggesting that there is a mechanistic link between cisplatin induced DNA damage, repair, and gene transcription regulation” does not logically follow from the data, which are simply PCA plots.

We agree that the logic does not follow here and have removed the sentence in the revised manuscript.

4. Line 43: “...were generated using isolated cell lines (including tumors)” – it is not clear what “including tumors” refers to when they are mentioning cell lines.

We have deleted “including tumors” in the revised manuscript.

5. The authors state “We didn’t find enrichment terms for circadian rhythm in liver and spleen. I think the authors are confused between genes involved in circadian rhythm and whether they are involved with “statistical enrichment” of the class of genes annotated with “circadian rhythm” as GO terms.

Thanks for spotting this! We have updated the corresponding section as follows: “We did not find statistically significant enrichment of genes listed under circadian rhythm in GO annotation in liver or spleen.”

6. The section on integration of the data with epigenetic markers (H3K4me1, H3K4me3, H3K27ac, H3K27me3, H3K36me3, POLR2A, and DNase I markers) is underwhelming. This section could be expanded upon.

Thanks. In this revision, we have carried out additional genome-wide analysis of DNA damage and repair, with association of different chromatin states. The chromatin states, shown in Figure 4c, are derived from histone posttranslational modifications and genomic sequence elements, with functional roles inferred of each state (Bogu, et al., 2015). We found that, similar to previous results (Hu, et al., 2016), DNA repair is higher in active chromatin states such as active promoters and CpG islands, the latter of which are enriched at transcription start

sites (Figure 4c). The distribution of the damage, on the other hand, with some minor differences, is rather uniform (Figure 4c). The differences in damage frequency that we observe can potentially be attributed to differences in DNA repair and number of GG dinucleotides.

7. There are a number of grammatical errors throughout the manuscript. A few are highlighted below:

- i) Line 36. Change “Reoccurence” to “recurrence”
- ii) Line 50: “...when extending the application to the in vivo situation.” Not clear what “situation refers to.”
- iii) Line 63-64: “Specifically, cisplatin was administered by intraperitoneal injection in mouse.”
- iv) Lines 175-176: “...circadian rhythm and rhythmic process were enrichment among the genes...”
- v) Lines 178-179: “We didn’t find enrichment terms for circadian rhythm in liver and spleen.” I believe that the authors are a bit confused between involvement of certain genes
- vi) Line 247-249: “For simplicity and without loss of generality, we focus on comparing liver against the other organs from this point on, while the results generalize.” This statement should be rewritten.
- vii) “...we further investigate how their excision repair and epigenomic profiles, with results shown in Fig 4.”

Thanks! In this revision, we have corrected all the grammatical errors as pointed above (changes are marked in the updated manuscript) and carried out careful proofread.

Reviewer #3:

The manuscript by Yimit et al generates damage and repair maps after 4 hours of cisplatin treatment in 4 mouse organs. Other cisplatin damage and repair maps have been published by the same lab although in cell lines (see Hu et al., PNAS 2016). The novelty of this paper relies in the in vivo characterization of the damage and repair and comparison between 4 different organs. Data is interesting and valuable, although some of the major conclusions obtained are confirmation of previous work on cell lines. Some questions and comments for the authors:

The protocol involves “Total RNA was extracted from mouse kidney, liver, lung and spleen” 4h after cisplatin treatment. However different organs have a different mixture of cell types that may have very different expression and epigenomic profiles, they may also have different tolerance to damage and different level of excision repair. While liver is quite homogeneous in terms of cell types within the organ, kidney is a very complex organ with many different cell types. Conclusions of the amount of damage and repair at the level of organ may mask differences within cell types. For example, a medium amount of excision repair, may be due to some cell types with very high level of repair and others cell types in the same organ with very low repair. How do authors deal with this? Some comments on the paper about this complexity would be useful.

Thanks very much for your thoughtful comments. Damage-seq and XR-seq were originally developed for studying damage formation and repair in tissue culture in vitro. To our best knowledge, this is the first study that adapts the technologies for use at whole-organism level. Cell-type specific DNA damage and repair within an organ are hard to assay. This is due to the technical and technological limitation in identifying constituent cell types in a heterogeneous tissue and gaining enough cell-type specific excision oligos before they get degraded. Therefore, we feel that the investigation of cell-type specific DNA damage and repair would be more suitable for a separate study in the future.

This is now explained in the Discussion section as follows: “It is noteworthy that the inherent heterogeneity of tissue samples from mouse kidney, liver, lung, and spleen makes our measurements potentially biased as attenuated means across different cell types. Cell-type specific DNA damage and repair within an organ are hard to assay. This is due to the technical and technological limitation in identifying constituent cell types in a

heterogeneous tissue and gaining enough cell-type specific excision oligos before they get degraded. As such, in this study we have resorted to “bulk” RNA-seq, XR-seq, and Damage-seq, while to develop high-throughput single-cell assay of DNA damage and repair is one of the future directions for this study and can be of great impact.”

Is the protocol of ENCODE to generate the epigenome data of the mouse comparable in terms on how and when RNA was extracted? Do they also take RNA from the whole organ? Are mice of similar age? If this were not the case, it does not invalidate the analysis, but would be interesting to inform in the paper about possible differences in the protocol if so, and what can they mean when interpreting the results.

Thanks for your thoughtful comments! In this revision, we have generated a new table as Supplementary Table S4, which includes the accession number as well as the age of mice for the ENCODE data that we utilized. We have also made further clarification in the Methods section under “ENCODE data” as follows:

“Epigenomic data across four mouse organs were downloaded from the ENCODE portal as bed files and visualized using the UCSC browser (Supplementary Table S4). All of the ENCODE data were generated from mice without cisplatin treatment, which enabled us to study DNA damage and repair within the context of pre-existing epigenomic profiles. While the effect of cisplatin treatment on epigenomic profiles would be of interest, here we focused on the genes that do not show significant up- or down-regulation after cisplatin treatment across all organs tested. In addition, most of the ENCODE data were generated using 8-week old young adult mice (Supplementary Table S4).”

Figure 1. Why XR-seq has not been done in control group?

Yes, we have performed XR-seq only in the treatment group. A single dose of cisplatin was administered to the cisplatin group and saline solution was administered to the control group. As shown in Supplementary Figure 2, we did not detect any damage enrichment (GG dinucleotides) in the control group. The specificity of XR-seq is such that background is virtually zero, and without nucleotide excision fragments it is impossible to perform XR-seq.

It seems, although it is not completely clear from the text, that all measurements are at the level of genes. Could authors clarify this? For example, in supplementary figure 15 and 16, each dot in the correlation plots is a gene? It is necessary that this is clarified in methods and figure legend. The damage and repair data are not restricted to genic regions, neither it is the epigenome data from ENCODE. Why the authors have restricted their analysis to genic regions? Couldn't they have analyzed the rest of the genome to make a more complete study?

Yes, each dot in Supplementary Figure S15-S16 (S16-S17 in current version) corresponds to a gene. We have explicitly pointed this out in the figure legends as well as in the Methods section. Furthermore, we have carried out genome-wide analysis of DNA damage and repair. Results are included in the newly generated Figure 4c and in the Results section under “Whole-genome analysis of DNA damage and repair.”

Figure 4c. Whole-genome analysis results of DNA damage and repair, with different genomic annotations. Analysis of repair (left) and damage (right) levels across nine genomic annotations for mouse liver reveals uniform distribution of damage but higher repair in active promotor, CpG island (enriched at transcription start sites), transcription elongation and transition regions in genome.

Page 6. “We then applied PCA to the normalized read count matrix and found that samples from the same organ are clustered together, indicating organ-specific DNA damage (Fig. 1b)”. That could be due to organ specific DNA repair during the 4 hours.

Thanks for pointing this out. PCA is applied to the read count matrix of the NTS reads from Damage-seq. As we show in Figure 3, the proportion of damage repaired by global repair in the NTS is negligible compared to the total damage induced. To be more precise, we updated the corresponding section in the revised manuscript as follows: “We then applied PCA to the normalized read count matrix from the NTS, where a negligible fraction of total DNA damage is repaired by GR only (Refer to Section “Effect of transcription on damage formation and repair” for more details). We found that samples from the same organ are clustered together, indicating organ-specific DNA damage at four hours (Fig.1b).”

Any idea, hypothesis on why spleen is a minor site for cisplatin damage, while kidney is a major site? Is this related to accessibility of the drug? Its metabolization?

As a body’s filter of blood, kidney accumulates cisplatin to a higher degree than other organs. The mechanism of cisplatin-induced kidney damage is complex and involves a number of interconnected factors, e.g., accumulation of cisplatin mediated by the transport pathway, DNA damage, and mitochondrial dysfunction. We assume that organ-specific cisplatin accumulation is related to accessibility of drugs. It is believed that approximately half of the initial platinum drug uptake depends on passive diffusion through cell membrane and that the other half depends on carrier-mediated transportation (Basu and Krishnamurthy, 2010). One of the predominant molecular mechanisms, by which kidney becomes the predominant cisplatin accumulation site, might be related to increased drug uptake through the cell membrane. We found that the expression of genes engaged in ion homeostasis significantly decreased after cisplatin treatment in kidney (Bsnd, Ptk2, Slc12a6, Slc12a7 and Tbxas1). BARTTIN (encoded by Bsnd gene) is a Cl⁻ channel β -subunit and plays an important role in renal salt reabsorption and potassium recycling in the inner ear (Estevez, et al., 2001). PTK2 is a non-receptor tyrosine kinase and loss of Ptk2 leads sensitize tumors to DNA-damaging chemotherapeutic agents in mouse model (Tavora, et al., 2014). SLC12A6 and SLC12A6 belong to the solute carrier family 12 and serve as a K⁺/Cl⁻ cotransporters in both kidney and ear (Lang, et al., 2007). TBXAC1 belongs to cytochrome P450 family and knock-out of Tbcass1 leads to hemostatic defect in mouse (Yu, et al., 2004). These previous reports and empirical

evidence suggest that cell membrane transporter alteration or changes in ion gradient within cytoplasm might be one of the reasons that kidney has higher cisplatin-induced DNA adducts and cisplatin-induced toxicity.

It is noteworthy that spleen is a minor site for cisplatin-induced DNA-adduct formation. In search for a potential mechanism, we noted that genes annotated with cell membrane trafficking, ferric acid transport, import into cell, membrane invagination, trivalent inorganic cation transport, vesicle-mediated transport, and immune system processes were down-regulated after cisplatin treatment. In addition, recent studies showed that spleen macrophages induce systematic chemo-resistance in vivo by secreting lysophospholipids (Houthuijzen, et al., 2014). It has also been reported that macrophages express drug-metabolizing cytochrome P450 superfamily enzymes and are involved in drug metabolism (Cataldi, et al., 2017). Overall, these data imply that spleen may remodel membrane transportation system to favor cisplatin resistance by reducing drug uptake or prevent cisplatin redistribution by repressing vesicle-mediated transport. Lastly, it also suggests that spleen may involve in cisplatin capture and metabolic process.

Above are results and hypotheses from our extensive literature search. With our direct evidence for support from our data, we wish not to include this in the main manuscript.

Is there any correlation between repair levels in different organs and expression of NER pathway genes?

Thanks for pointing this out. We did not find upregulation of genes involved in the DNA damage response and, in particular, the nuclear excision repair pathway, across all organs tested after cisplatin treatment. Our data suggest that the cisplatin-induced DNA damage efficiency and the alteration in the repair of the DNA damage could not be attributed to the changes in expression level of excision proteins at this stage. However, we cannot rule out a possibility that excision repair proteins make a cellular localization change instead of an increase in abundance in response to cisplatin-induced DNA damage. Several lines of indirect evidence support this hypothesis: cisplatin treatment of rat spiral ganglion neurons enables induction of nuclear localization of XPC (Xeroderma pigmentosum complementation group-C) and XPA (Xeroderma pigmentosum complementation group-A) without shifting the total Xpc and Xpa expression level (Guthrie, et al., 2008). Microscope-based studies in vitro have shown that XPC and XPA rapidly translocate from cytoplasm to nucleus in response to UV-induced DNA lesion (Hoogstraten, et al., 2008; Wu, et al., 2007). Importantly, recent data from our lab show that Xpa mRNA exhibits circadian rhythm in liver, and that repair of cisplatin-induced DNA adducts in kidney and liver are under control by the circadian programs (Kang, et al., 2010).

Supplementary figure 7, in spleen the mitochondrial normalized damage is equivalent to the non-mitochondrial, why is that?

It is an observation that we made and we currently do not have direct explanation for it. It is possible that the number of mitochondria or the number of mitochondrial DNA molecules per mitochondria are different in the spleen. We have no reference studies on this subject and thus do not wish to speculate.

“5.4% of reads mapping to mitochondrial DNA contain GG, which indicates that there is no excision repair”. What the other reads mean in an XR-seq experiment? Noise?

Yes, it is noise. This has been clarified in the revised manuscript.

Seems that expression analysis also shows a lower impact of cisplatin in spleen compared to other tissues (i.e. only 193 genes differentially expressed), consistent with lower amount of damage. Why is that? Accessibility of the drug to the organ?

Presumably yes. We agree that kidney is a major site of cisplatin DNA-adducts and has higher number of significantly up- and down-regulated genes, while spleen is a minor site for damage accumulation and has lower

number of significantly up- and down-regulated genes. We have offered a few hypotheses and reasonings for this observation in our response to the previous comment. We assume that organ-specific accessibility of drugs might be a factor that causes organ-specific transcription alteration but not the only mechanism.

The authors say “We did not observe significant correlations between the damage level in the NTS and the epigenetic markers, indicating a relatively uniform damage distribution along the genome”. But looking at supplementary figure 16 one of the strongest correlations (anticorrelation in this case) is between Damage (TS and NTS) and H3K36me (-0.46 and -0.46 in kidney). How do authors explain this?

This is one observation we made in our data and is only seen in kidney (positive correlation in liver and insignificant correlation in lung and spleen). The fact that the significant anticorrelation is only for H3K36me3 and that it does not generalize across the other organs indicates a possible spurious correlation in kidney. To be more precise, we have updated the text as “*We did not observe significant correlations, **in the same directions across all organs**, between the damage level in the NTS and the epigenetic markers. This **further supports the fact that damage is relatively uniformly distributed along the genome (Supplementary Figure S17).***”

Figure 4. How many genes are considered as “highly expressed in liver” in the analysis of Figure 4a, and how many as “weakly expressed in liver” in Figure 4b? This should be clarified.

602 and 414 genes are significantly highly and lowly expressed in liver compared to the other organs with absolute log fold change greater than 2, respectively. We have updated this in the revision.

References

- Aston, W.J., *et al.* A systematic investigation of the maximum tolerated dose of cytotoxic chemotherapy with and without supportive care in mice. *BMC Cancer* 2017;17(1):684.
- Bogu, G.K., *et al.* Chromatin and RNA Maps Reveal Regulatory Long Noncoding RNAs in Mouse. *Mol Cell Biol* 2015;36(5):809-819.
- Cataldi, M., *et al.* Emerging Role of the Spleen in the Pharmacokinetics of Monoclonal Antibodies, Nanoparticles and Exosomes. *Int J Mol Sci* 2017;18(6).
- Dakup, P.P., *et al.* The circadian clock regulates cisplatin-induced toxicity and tumor regression in melanoma mouse and human models. *Oncotarget* 2018;9(18):14524-14538.
- Estevez, R., *et al.* Barttin is a Cl⁻ channel beta-subunit crucial for renal Cl⁻ reabsorption and inner ear K⁺ secretion. *Nature* 2001;414(6863):558-561.
- Fichtinger-Schepman, A.M., *et al.* Platinum concentrations and DNA adduct levels in tumors and organs of cisplatin-treated LOU/M rats inoculated with cisplatin-sensitive or -resistant immunoglobulin M immunocytoma. *Cancer Res* 1989;49(11):2862-2867.
- Guthrie, O.W., *et al.* Cisplatin induces cytoplasmic to nuclear translocation of nucleotide excision repair factors among spiral ganglion neurons. *Hear Res* 2008;239(1-2):79-91.
- Hoogstraten, D., *et al.* Versatile DNA damage detection by the global genome nucleotide excision repair protein XPC. *J Cell Sci* 2008;121(Pt 17):2850-2859.
- Houthuijzen, J.M., *et al.* Lysophospholipids secreted by splenic macrophages induce chemotherapy resistance via interference with the DNA damage response. *Nat Commun* 2014;5:5275.
- Hu, J., *et al.* Cisplatin DNA damage and repair maps of the human genome at single-nucleotide resolution. *Proc Natl Acad Sci U S A* 2016;113(41):11507-11512.

- Kang, T.H., *et al.* Circadian control of XPA and excision repair of cisplatin-DNA damage by cryptochrome and HERC2 ubiquitin ligase. *Proc Natl Acad Sci U S A* 2010;107(11):4890-4895.
- Lang, F., *et al.* Functional significance of channels and transporters expressed in the inner ear and kidney. *Am J Physiol Cell Physiol* 2007;293(4):C1187-1208.
- Liedert, B., *et al.* Adduct-specific monoclonal antibodies for the measurement of cisplatin-induced DNA lesions in individual cell nuclei. *Nucleic Acids Res* 2006;34(6):e47.
- Tavora, B., *et al.* Endothelial-cell FAK targeting sensitizes tumours to DNA-damaging therapy. *Nature* 2014;514(7520):112-116.
- Wu, X., *et al.* ATR-dependent checkpoint modulates XPA nuclear import in response to UV irradiation. *Oncogene* 2007;26(5):757-764.
- Yu, I.S., *et al.* TXAS-deleted mice exhibit normal thrombopoiesis, defective hemostasis, and resistance to arachidonate-induced death. *Blood* 2004;104(1):135-142.

REVIEWERS' COMMENTS:

Reviewer #1 (Remarks to the Author):

The authors have generally replied to my comments in the rebuttal. However I have a few remaining comments that I think need addressed in the manuscript.

1. The authors need to comment in the manuscript on the tolerability of 10mg/Kg in the mice used in the study.
2. The authors need to comment in the manuscript whether they are measuring platinum momo-adducts.

Reviewer #3 (Remarks to the Author):

The authors have satisfactorily addressed all comments.

We thank all of the reviewers who have read and commented on our revised manuscript. Here are our point-by-point responses to the specific points.

REVIEWERS' COMMENTS:

Reviewer #1 (Remarks to the Author):

The authors have generally replied to my comments in the rebuttal. However, I have a few remaining comments that I think need addressed in the manuscript.

1. The authors need to comment in the manuscript on the tolerability of 10mg/Kg in the mice used in the study.

As requested, we have included a comment about tolerability of 10mg/Kg cisplatin in the C57BL/6J mice in the methods section.

2. The authors need to comment in the manuscript whether they are measuring platinum mono-adducts.

We appreciate the reviewer's question. However, mono-adducts constitute only 2-3% of cisplatin adducts (Elizabeth, et al., 1999) and our Damage-seq and XR-seq methods do not have the resolution to measure such a low frequency event. We wish not to discuss this issue at any length in our manuscript so as not to distract from the main point of the paper that deals with Pt-d(GpG) di-adduct, which are accepted by the scientific community as the primary toxic adducts of cisplatin.

Reviewer #3 (Remarks to the Author):

The authors have satisfactorily addressed all comments.

Thank you. We highly appreciate it.
No comments to be addressed.

References

Elizabeth, R., *et al.* Structure, recognition, and processing of cisplatin-DNA adducts. *Chem. Rev.* 1999; 99(9): 2467-2498